# JECC: Commonsense Reasoning Tasks Derived from Interactive Fictions

**Mo Yu**[1]  **Xiaoxiao Guo**[1]  **Yufei Feng**[2]  **Yi Gu**[1]
**Xiaodan Zhu**[2]  **Michael Greenspan**[2]  **Murray Campbell**[1]  **Chuang Gan**[1]
[1] IBM Research  [2] Queens University
yum@us.ibm.com  xiaoxiao.guo@ibm.com  feng.yufei@queensu.ca

## Abstract

Commonsense reasoning simulates the human ability to make presumptions about our physical world, and it is an essential cornerstone in building general AI systems. We propose a new commonsense reasoning dataset based on human's Interactive Fiction (IF) gameplay walkthroughs as human players demonstrate plentiful and diverse commonsense reasoning. The new dataset provides a natural mixture of various reasoning types and requires multi-hop reasoning. Moreover, the IF game-based construction procedure requires much less human interventions than previous ones. Experiments show that the introduced dataset is challenging to previous machine reading models with a significant 20% performance gap compared to human experts.

## 1 Introduction

There has been a flurry of datasets and benchmarks proposed to address natural language-based commonsense reasoning [11, 27, 20, 13, 9, 15, 2, 8, 3, 16, 26]. These benchmarks usually adopt a multi-choice form – with the input query and an optional short paragraph of the background description, each candidate forms a statement; the task is to predict the statement that is consistent with some commonsense knowledge facts.

These benchmarks share some limitations, as they are mostly constructed to focus on a single reasoning type and require similar validation-based reasoning. First, most benchmarks concentrate on a specific facet and ask human annotators to write candidate statements related to the particular type of commonsense. As a result, the distribution of these datasets is unnatural and biased to a specific facet. For example, most benchmarks focus on collocation, association, or other relations (e.g., ConceptNet [18] relations) between words or concepts [11, 20, 13, 9]. Other examples include temporal commonsense [27], physical interactions between actions and objects [3], emotions and behaviors of people under the given situation [16], and cause-effects between events and states [15, 2, 8]. Second, most datasets require validation-based reasoning between a commonsense fact and a text statement but neglect hops over multiple facts. [1] The previous work's limitations bias the model evaluation. For example, pre-trained Language Models (LMs), such as BERT [4], well handled most benchmarks. Their pre-training process may include texts on the required facts, enabling adaptation to the dominating portion of commonsense validation instances. The powerful LMs with sufficient capacity can fit the isolated reasoning type easily. As a result, the above limitations of previous benchmarks lead to discrepancies among practical NLP tasks that require broad reasoning ability on various facets.

---

[1] Some datasets include a portion of instances that require explicit reasoning capacity, such as [2, 8, 3, 16]. But still, standalone facts can solve most such instances.

Submitted to the 35th Conference on Neural Information Processing Systems (NeurIPS 2021) Track on Datasets and Benchmarks. Do not distribute.

**Our Contribution.** We derive *a new commonsense reasoning dataset from the model-based reinforcement learning challenge* of Interactive Fictions (IF) to address the above limitations. Recent advances [7, 1, 5] in IF games have recognized several commonsense reasoning challenges, such as detecting valid actions and predicting different actions' effects. Figure 1 illustrates sample gameplay of the classic game *Zork1* and the required commonsense knowledge. We derive a commonsense dataset from human players' gameplay records related to the second challenge, i.e., predicting which textual observation is most likely after applying an action or a sequence of actions to a given game state.

The derived dataset naturally addresses the aforementioned limitations in previous datasets. First, predicting the next observation naturally requires various commonsense knowledge and reasoning types. As shown in Figure 1, a primary commonsense type is spatial reasoning, e.g., ''climb the tree'' makes the protagonist up on a tree. Another primary type is reasoning about object interactions. For example, keys can open locks (object relationships); ''hatch egg'' will reveal "things" inside the egg (object properties); ''burn repellent with torch'' leads to an explosion and kills the player (physical reasoning). The above interactions are more comprehensive than the relationships defined in ConceptNet as used in previous datasets. Second, the rich textual observation enables more complex reasoning over direct commonsense validation. Due to the textual observation's narrative nature, a large portion of the textual observations are not a sole statement of the action effect, but an extended narrates about what happens because of the effect.[2] Third, our commonsense reasoning task formulation shares the essence of dynamics model learning for model-based RL solutions related to world models and MuZero [6, 17]. As a result, models developed on our benchmarks provide direct values to model-based reinforcement learning for text-game playing.

Finally, compared to previous works that heavily rely on human annotation, our dataset construction requires minimal human effort, providing great **expansibility** to our dataset. For example, with large amounts of available IF games in dungeon crawls, Sci-Fi, mystery, comedy, and horror, it is straightforward to extend our dataset to include more data samples and cover a wide range of genres. We can also naturally increase the reasoning difficulty by increasing the prediction horizon of future observations after taking multi-step actions instead of a single one.

In summary, we introduce a new commonsense reasoning dataset construction paradigm, collectively with two datasets. The larger dataset covers 29 games in multiple domains from the *Jericho Environment* [7], named the Jericho Environment Commonsense Comprehension task (**JECC**). The smaller dataset, aimed for the single-domain test and fast model development, includes four IF games in the *Zork Universe*, named Zork Universe Commonsense Comprehension (**ZUCC**). We provide strong baselines to the datasets and categorize their performance gap compared to human experts.

Figure 1: Classic dungeon game *Zork1* gameplay sample. The player receives textual observations describing the current game state and sends textual action commands to control the protagonist. Various commonsense reasoning is illustrated in the texts of observations and commands from the gameplay interaction, such as spatial relations, objective manipulation, and physical relations.

---

[2]For some actions, such as `get` and `drop` objects, the next observations are simple statements. We removed some of these actions. Details can be found in Section 3.

## 2    Related Work

Previous work has identified various types of commonsense knowledge humans master for text understanding. As discussed in the introduction section, most existing datasets cover one or a few limited types. Also, they mostly have the form of validation between a commonsense knowledge fact and a text statement.

**Semantic Relations between Concepts.** Most previous datasets cover the semantic relations between words or concepts. These relations include the concept hierarchies, such as those covered by WordNet or ConceptNet, and word collocations and associations. For example, the early work Winograd [11] evaluates the model's ability to capture word collocations, associations between objects, and their attributes as a pronoun resolution task. The work by [20] is one of the first datasets covering the ConceptNet relational tuple validation as a question-answering task. The problem asks the relation of a source object, and the model selects the target object that satisfies the relation from four candidates. [13] focus on the collocations between adjectives and objects. Their task takes the form of textual inference, where a premise describes an object and the corresponding hypothesis consists of the object that is modified by an adjective. [9] study associations among multiple words, i.e., whether a word can be associated with two or more given others (but the work does not formally define the types of associations). They propose a new task format in games where the player produces as many words as possible by combining existing words.

**Causes/Effects between Events or States.** Previous work proposes datasets that require causal knowledge between events and states [15, 2, 8]. [15] takes a text generation or inference form between a cause and an effect. [2] takes a similar form to ours – a sequence of two observations is given, and the model selects the plausible hypothesis from multiple candidates. Their idea of data construction can also be applied to include any types of knowledge. However, their dataset only focuses on causal relations between events. The work of [8] utilizes multi-choice QA on a background paragraph, which covers a wider range of casual knowledge for both events and statements.

**Other Commonsense Datasets.** [27] proposed a unique temporal commonsense dataset. The task is to predict a follow-up event's duration or frequency, given a short paragraph describing an event. [3] focus on physical interactions between actions and objects, namely whether an action over an object leads to a target effect in the physical world. These datasets can be solved by mostly applying the correct commonsense facts; thus, they do not require reasoning. [16] propose a task of inferring people's emotions and behaviors under the given situation. Compared to the others, this task contains a larger portion of instances that require reasoning beyond fact validation. The above tasks take the multi-choice question-answering form.

**Next-Sentence Prediction.** The next sentence prediction tasks, such as SWAG [26], are also related to our work. These tasks naturally cover various types of commonsense knowledge and sometimes require reasoning. The issue is that the way they guarantee distractor candidates to be irrelevant greatly simplified the task. In comparison, our task utilizes the IF game engine to ensure actions uniquely determining the candidates, and ours has human-written texts.

Finally, our idea is closely related to [25], which creates a task of predicting valid actions for each IF game state. [25, 24] also discussed the advantages of the supervised tasks derived from IF games for natural langauge understanding purpose.

## 3    Dataset Construction: Commonsense Challenges from IF Games

We pick games supported by the *Jericho* environment [7] to construct the **JECC** dataset.[3] We pick games in the *Zork Universe* for the **ZUCC** dataset.[4] We first introduce the necessary definitions in the IF game domain and then describe how we construct our **ZUCC** and **JECC** datasets as the forward prediction tasks based on human players' gameplay records, followed by a summary on the improved properties of our dataset compared to previous ones. The dataset will be released for public usage. It can be created with our released code with MIT License.

---

[3]We collect the games *905, acorncourt, advent, adventureland, afflicted, awaken, balances, deephome, dragon, enchanter, inhumane, library, moonlit, omniquest, pentari, reverb, snacktime, sorcerer, zork1* for training, *zork3, detective, ztuu, jewel, zork2* as the development set, *temple, gold, karn, zenon, wishbringer* as the test set.

[4]We pick *Zork1, Enchanter*, and *Sorcerer* as the training set, and the dev and sets are non-overlapping split from *Zork3*.

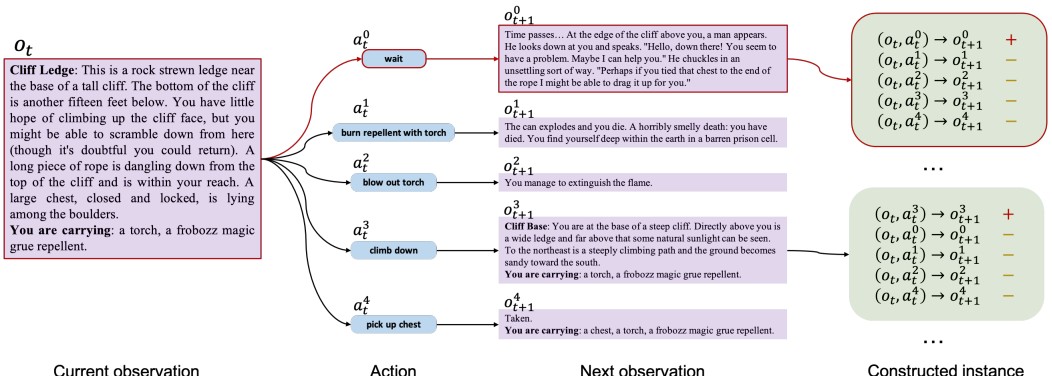

Figure 2: Illustration of our data construction process, taking an example from *Zork3*. $+/-$: positive/negative labels. The red colored path denotes the tuple and the resulted data instance from the human walkthrough.

Table 1: Data statistics of our **ZUCC** and **JECC** tasks. **WT** stands for walkthrough. The evaluation sets of **JECC** only consist of tuples in walkthroughs. The evaluation sets of **ZUCC** consist of all tuples after post-processing. For **JECC** the total numbers of tuples in the training games and evaluation games are close. Yet as discussed in the dataset construction criteria (Section 3.3), we only evaluate the models with tuples from the walkthroughs to ensure a representative distribution of required knowledge.

| | #WT Tuples | #Tuples before Proc | #Tuples after Proc |
|---|---|---|---|
| **ZUCC** | | | |
| Train | 913 | 17,741 | 10,498 |
| All Eval | 271 | 4,069 | 2,098 |
| Dev | – | – | 1,276 |
| Test | – | – | 822 |
| **JECC** | | | |
| Train | 2,526 | 48,843 | 24,801 |
| All Eval | 2,063 | 53,160 | 25,891 |
| Dev | 917 | – | – |
| Test | 1,146 | – | – |

## 3.1 Interactive Fiction Game Background

Each IF game can be defined as a Partially Observable Markov Decision Process (POMDP), namely a 7-tuple of $\langle S, A, T, O, \Omega, R, \gamma \rangle$, representing the hidden game state set, the action set, the state transition function, the set of textual observations composed from vocabulary words, the textual observation function, the reward function, and the discount factor respectively. The game playing agent interacts with the game engine in multiple turns until the game is over or the maximum number of steps is reached. At the $t$-th turn, the agent receives a textual observation describing the current game state $o_t \in O$ and sends a textual action command $a_t \in A$ back. The agent receives additional reward scalar $r_t$ which encodes the game designers' objective of game progress. Thus the task of the game playing can be formulated to generate a textual action command per step as to maximize the expected cumulative discounted rewards $\mathbf{E}\left[\sum_{t=0}^{\infty} \gamma^t r_t\right]$. Most IF games have a deterministic dynamics, and the next textual observation is uniquely determined by an action choice. Unlike most previous work on IF games that design autonomous learning agents, we utilize human players' gameplay records that achieve the highest possible game scores.

**Trajectories and Walkthroughs.** A *trajectory* in text game playing is a sequence of tuples $\{(o_t, a_t, r_t, o_{t+1})\}_{t=0}^{T-1}$, starting with the initial textual observation $o_0$ and the game terminates at time step $t = T$, i.e., the last textual observation $o_T$ describes the game termination scenario. We define the *walkthrough* of a text game as a trajectory that completes the game progress and achieves the highest possible game scores.

## 3.2 Data Construction from the Forward Prediction Task

**The Forward Prediction Task.** We represent our commonsense reasoning benchmark as a next-observation prediction task, given the current observation and action. The benchmark construction starts with all the tuples in a walkthrough trajectory, and we then extend the tuple set by including all valid actions and their corresponding next-observations conditioned on the current observations in the walkthrough. Specifically, for a walkthrough tuple $(o_t, a_t, r_t, o_{t+1},)$, we first obtain the complete valid action set $A_t$ for $o_t$. We sample and collect one next observation $o_{t+1}^j$ after executing the corresponding action $a_t^j \in A_t$. The next-observation prediction task is thus to select the next observation $o_{t+1}^j$ given $(o_t, a_t^j)$ from the complete set of next observations $O_{t+1} = \{o_{t+1}^k, \forall k\}$. Figure 2 illustrates our data construction process.

**Data Processing.** We collect tuples from the walkthrough data provided by the Jericho environments. We detect the valid actions via the Jericho API and the game-specific templates. Following previous work [7], we augmented the observation with the textual feedback returned by the command [*inventory*] and [*look*]. The former returns the protagonist's objects, and the latter returns the current location description. When multiple actions lead to the same next-observation, we randomly keep one action and next-observation in our dataset. We remove the `drop OBJ` actions since it only leads to synthetic observations with minimal variety. For each step $t$, we keep at most 15 candidate observations in $O_t$ for the evaluation sets. When there are more than 15 candidates, we select the candidate that differs most from $o_t$ with Rouge-L measure [12].

During evaluation, for **JECC**, we only evaluate on the tuples on walkthroughs. As will be discussed in 3.3, this helps our evaluation reflects a natural distribution of commonsense knowledge required, which is an important criterion pointed out by our introduction. However for **ZUCC** the walkthough data is too small, therefore we consider all the tuples during evaluation. This leads to some problems. First, there are actions that do not have the form of `drop OBJ` but have the actual effects of dropping objects. Through the game playing process, more objects will be collected in the inventory at the later stages. These cases become much easier as long as these non-standard drop actions have been recognized. A similar problem happens to actions like `burn repellent` that can be performed at every step once the object is in the inventory. To deal with such problems, we down-sample these biased actions to achieve similar distributions in development and test sets. Table 1 summarizes statistics of the resulted **JECC** and **ZUCC** datasets.

## 3.3 Design Criterion and Dataset Properties

**Knowledge coverage and distribution.** As discussed in the introduction, an ideal commonsense reasoning dataset needs to cover various commonsense knowledge types, especially useful ones for understanding language. A closely related criterion is that the required commonsense knowledge and reasoning types should reflect a natural distribution in real-world human language activities.

Our **JECC** and **ZUCC** datasets naturally meet these two criteria. The various IF games cover diverse domains, and human players demonstrate plentiful and diverse commonsense reasoning in finishing the games. The commonsense background information and interventions are recorded in human-written texts (by the game designers and the players, respectively). With the improved coverage of commonsense knowledge following a natural distribution, our datasets have the potential of better evaluating reasoning models, alleviating the biases from previous datasets on a specific knowledge reasoning type.

**Reasoning beyond verification.** A reasoning dataset should evaluate the models' capabilities in (multi-hop) reasoning with commonsense facts and background texts, beyond simple validation of knowledge facts.

By design, our datasets depart from simple commonsense validation. Neither the input (current observation and action) nor the output (next observation) directly describes a knowledge fact. Instead, they are narratives that form a whole story. Moreover, our task formulation explicitly requires using commonsense knowledge to understand how the action impacts the current state, then reason the effects, and finally verifies whether the next observation coheres with the action effects. These solution steps form a multi-step reasoning process.

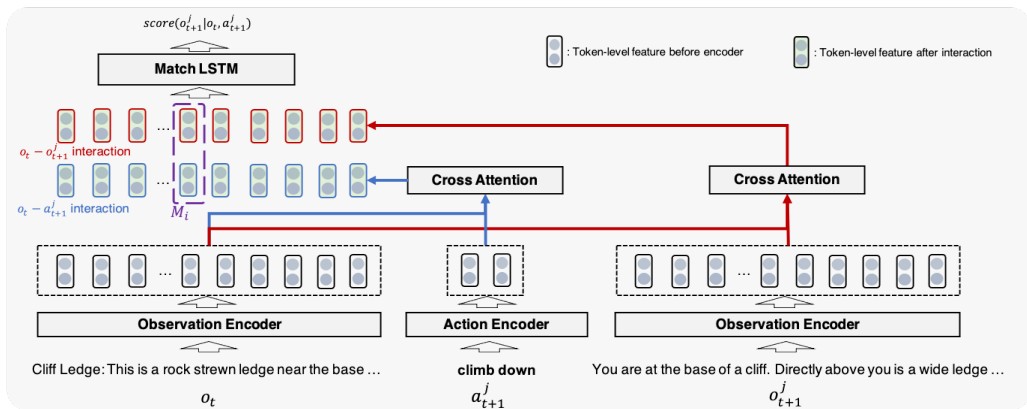

Figure 3: The co-matching architecture for our tasks.

**Limitations** Our dataset construction method has certain limitations. One important limitation is that it is difficult to get the distribution of the required commonsense knowledge types. This can be addressed in future work with human designed commonsense knowledge schema and human annotation.

## 4 Neural Inference Baselines

We formulate our task as a textual entailment task that the models infer the next state $o_{t+1}$ given $o_t$ and $a_t$. We provide strong textual entailment-based baselines for our benchmark. We categorize the baselines into two types, namely pairwise textual inference methods and the triplewise inference methods. The notations $o_t$, $a_t$ of observations and actions represent their word sequences.

### 4.1 Neural Inference over Textual Pairs

• **Match LSTM** [22] represents a commonly used natural language inference model. Specifically, we concatenate $o_t$ and $a_t$ separated by a special split token as the premise and use the $o_{t+1}^j$, $j = 1, ...N$ as the hypothesis. For simplicity *we denote $o_t$, $a_t$ and a candidate $o_{t+1}^j$ as $o, a, \tilde{o}$.* We encode the premise and the hypothesis with bidirectional-LSTM model:

$$\boldsymbol{H}^{o,a} = \text{BiLSTM}([o, a]), \boldsymbol{H}^{\tilde{o}} = \text{BiLSTM}(\tilde{o}), \quad (1)$$

where $\boldsymbol{H}^{o,a}$ and $\boldsymbol{H}^{\tilde{o}}$ are the sequences of BiLSTM output $d$-dimensional hidden vectors that correspond to the premise and hypothesis respectively. We apply the bi-attention model to compute the match between the premise and the hypothesis, which is followed by a Bi-LSTM model to get the final hidden sequence for prediction:

$$\bar{\boldsymbol{H}}^{\tilde{o}} = \boldsymbol{H}^{\tilde{o}}\boldsymbol{G}^{\tilde{o}}, \boldsymbol{G}^{\tilde{o}} = \text{SoftMax}((W^g\boldsymbol{H}^{\tilde{o}} + b^g \otimes e)^T \boldsymbol{H}^{o,a})$$

$$\boldsymbol{M} = \text{BiLSTM}([\boldsymbol{H}^{o,a}, \bar{\boldsymbol{H}}^{\tilde{o}}, \boldsymbol{H}^{o,a} - \bar{\boldsymbol{H}}^{\tilde{o}}, \boldsymbol{H}^{o,a} \odot \bar{\boldsymbol{H}}^{\tilde{o}}]).$$

Here $W^g \in \mathbb{R}^{d \times d}$ and $b^g \in \mathbb{R}^d$ are learnable parameters and $e \in \mathbb{R}^{|\tilde{o}|}$ denotes a vector of all 1s. We use a scoring function $f(\cdot)$ to compute matching scores of the premise and the hypothesis via a linear transformation on the max-pooled output of $\boldsymbol{M}$. The matching scores for all $\tilde{o}$ are then fed to a softmax layer for the final prediction. We use the cross-entropy loss as the training objective.

• **BERT Siamese** uses a pre-trained BERT model to separately encode the current observation-action pair $(o_t, a_t)$ and candidate observations $\tilde{o}$. All inputs to BERT start with the "[CLS]" token, and we concatenate $o_t$ and $a_t$ with a "[SEP]" token:

$$\boldsymbol{h}^{o,a} = \text{BERT}([o, a]), \quad \boldsymbol{h}^{\tilde{o}} = \text{BERT}(\tilde{o}),$$

$$l_j = f([\boldsymbol{h}^{o,a}, \boldsymbol{h}^{\tilde{o}}, \boldsymbol{h}^{o,a} - \boldsymbol{h}^{\tilde{o}}, \boldsymbol{h}^{o,a} \odot \boldsymbol{h}^{\tilde{o}}]),$$

where $[\cdot, \cdot]$ denotes concatenation. $\boldsymbol{h}^{o,a}$ and $\boldsymbol{h}^{\tilde{o}}$ are the last layer hidden state vectors of the "[CLS]" token. Similarly, the scoring function $f$ computes matching scores for candidate next-observations

by linearly projecting the concatenated vector into a scalar. The matching scores of all $\tilde{o}$ are grouped to a softmax layer for the final prediction.

- **BERT Concat** represents the standard pairwise prediction mode of BERT. We concatenate $o$ and $a$ with a special split token as the first segment and treat $\tilde{o}$ as the second. We then concatenate the two with the "[SEP]" token:

$$l_j = f(\text{BERT}([o, a, \tilde{o}])).$$

The scoring function $f$ linearly projects the last-layer hidden state of the "[CLS]" token into a scalar, and the scores are grouped to a softmax layer for final prediction. This model is much less efficient than the former two as it requires explicit combination of observation-action-next-observation as inputs. Thus this model is impractical due to the huge combinatorial space. Here we report its results for reference.

## 4.2 Neural Inference over Textual Triples

Existing work [10, 19, 21] has applied textual matching and entailment among triples. For example, when applying to multi-choice QA, the entailment among triples is to predict whether a question $q$, an answer option $a$ can be supported by a paragraph $p$. In this section, we apply the most commonly used co-matching approaches [23] and its BERT variant to our task. Figure 3 illustrates our co-matching architecture.

Table 2: Evaluation on our datasets. Human performance (*) is computed on subsets of our data. BERT-concat (†) performs not well on JECC dev set, because the dev instances are longer on average. The concatenated inputs are more likely beyond BERT's length limit. **Inference speeds** of models are evaluated on the development set of our **JECC** dataset with a single V100 GPU.

| Method | ZUCC | | JECC | | Inference Speed (#states/sec) | #Parameters |
|---|---|---|---|---|---|---|
| | Dev Acc | Test Acc | Dev Acc | Test Acc | | |
| Random Guess | 10.66 | 16.42 | 7.92 | 8.01 | – | – |
| *Textual Entailment Baselines* | | | | | | |
| Match LSTM | 57.52 | 62.17 | 64.99 | 66.14 | 33.8 | 1.43M |
| BERT-siamese | 49.29 | 53.77 | 61.94 | 63.87 | 9.1 | 109.49M |
| BERT-concat | 64.73 | 64.48 | 67.39† | 72.16 | 0.6 | 109.48M |
| *Triple Modeling Baselines* | | | | | | |
| Co-Match LSTM | 72.34 | 75.91 | 70.01 | 71.64 | 25.8 | 1.47M |
| Co-Match BERT | 72.79 | 75.56 | 74.37 | 75.48 | 7.0 | 110.23M |
| Human Performance* | 96.40 | – | 92.0 | – | – | – |

- **Co-Matching LSTM** [23] jointly encodes the question and answer with the context passage. We extend the idea to conduct the multi-hop reasoning in our setup. Specifically, similar to Equation 1, we first encode the current state observation $o$, the action $a$ and the candidate next state observation $\tilde{o}$ separately with a BiLSTM model, and use $\boldsymbol{H}^o, \boldsymbol{H}^a, \boldsymbol{H}^{\tilde{o}}$ to denote the output hidden vectors respectively.

We then integrate the co-matching to the baseline readers by applying bi-attention described in Equation 2 on $(\boldsymbol{H}^o, \boldsymbol{H}^{\tilde{o}})$, and $(\boldsymbol{H}^a, \boldsymbol{H}^{\tilde{o}})$ using the same set of parameters:

$$\bar{\boldsymbol{H}}^o = \boldsymbol{H}^o \boldsymbol{G}^o, \boldsymbol{G}^o = \text{SoftMax}((W^g \boldsymbol{H}^o + b^g \otimes e_o)^T \boldsymbol{H}^{\tilde{o}})$$
$$\bar{\boldsymbol{H}}^a = \boldsymbol{H}^a \boldsymbol{G}^a, \boldsymbol{G}^a = \text{SoftMax}((W^g \boldsymbol{H}^a + b^g \otimes e_a)^T \boldsymbol{H}^{\tilde{o}}),$$

where $W^g \in \mathbb{R}^{d \times d}$ and $b^g \in \mathbb{R}^d$ are learnable parameters and $e_o \in \mathbb{R}^{|o|}, e_a \in \mathbb{R}^{|a|}$ denote vectors of all 1s. We further concatenate the two output hidden sequences $\bar{\boldsymbol{H}}^o$ and $\bar{\boldsymbol{H}}^a$, followed by a BiLSTM model to get the final sequence representation:

$$\boldsymbol{M} = \text{BiLSTM}(\begin{bmatrix} \boldsymbol{H}^{\tilde{o}}, \bar{\boldsymbol{H}}^o, \boldsymbol{H}^{\tilde{o}} - \bar{\boldsymbol{H}}^o, \boldsymbol{H}^{\tilde{o}} \odot \bar{\boldsymbol{H}}^o \\ \boldsymbol{H}^{\tilde{o}}, \bar{\boldsymbol{H}}^a, \boldsymbol{H}^{\tilde{o}} - \bar{\boldsymbol{H}}^a, \boldsymbol{H}^{\tilde{o}} \odot \bar{\boldsymbol{H}}^a \end{bmatrix}) \tag{2}$$

A scoring function $f$ linearly projects the max-pooled output of $\boldsymbol{M}$ into a scalar.

259 • **Co-Matching BERT** replaces the LSTM encoders with BERT encoders. Specifically, it separately
260 encodes $o, a, \tilde{o}$ with BERT. Given the encoded hidden vector sequences $\boldsymbol{H}^o$, $\boldsymbol{H}^a$ and $\boldsymbol{H}^{\tilde{o}}$, it follows
261 Co-Matching LSTM's bi-attention and scoring function to compute the matching score.

## 5 Experiments

263 We first evaluate all the proposed baselines on our datasets. Then we conduct a human study on a
264 subset of our development data to investigate how human experts perform and the performance gap
265 between machines and humans.

266 **Implementation Details.**   We set learning rate of Adam to $1\mathrm{e}^{-3}$ for LSTM-based models and $2\mathrm{e}^{-5}$
267 for BERT-based models. The batch size various, each corresponds to the number of valid actions
268 (up to 16 as described in data construction section). For the LSTM-based models, we use the Glove
269 embedding [14] with 100 dimensions. For both match LSTM, co-match LSTM and co-match BERT,
270 we map the final matching states $M$ to 400 dimensional vectors, and pass these vectors to a final
271 bi-directional LSTM layer with 100-dimensional hidden states.

272 All the experiments run on servers using a single Tesla V100 GPU with 32G memory for both training
273 and evaluation. We use Pytorch 1.4.0; CUDA 10.2; Transformer 3.0.2; and Jericho 2.4.3.

### 5.1  Overall Results

275 Table 2 summarizes the models' accuracy on the development and test splits and the inference
276 speed on the **JECC** development set.  First, all the baselines learned decent models, achieving
277 significantly better scores than a random guess.  Second, the co-matching ones outperform their
278 pairwise counterparts (Co-Match BERT > BERT-Siamese/-Concat, Co-Match LSTM > Match LSTM),
279 and the co-match BERT performs consistently best on both datasets. The co-matching mechanism
280 better addressed our datasets' underlying reasoning tasks, with a mild cost of additional inference
281 computation overhead.  Third, the co-match LSTM well balances accuracy and speed. In contrast, the
282 BERT-concat, although still competitive on the accuracy, suffers from a quadratic time complexity on
283 sequence lengths, prohibiting practical model learning and inference.

284 BERT-Concat represents recent general approaches to commonsense reasoning tasks. We manually
285 examined the incorrect predictions and identified two error sources. First, it is challenging for the
286 models to distinguish the structures of current/next observations and actions, especially when directly
287 taking as input complicated concatenated strings of multiple types of elements. For example, it may
288 not learn which parts of the inputs correspond to inventories. Second, the concatenation often makes
289 the texts too long for BERT.

290 Albeit the models consistently outperform random guesses, the best development results on both
291 datasets are still far below human-level performance. Compared to the human expert's near-perfect
292 performance, the substantial performance gaps confirm that our datasets require important common-
293 sense that humans always possess.

294 **Remark on the Performance Consistency.**   It seems that the BERT-Concat and co-match
295 LSTM/BERT models achieve inconsistent results on the **ZUCC** and **JECC**. We point out that
296 this inconsistency is mainly due to the different distributions – for the **JECC** we hope to keep a
297 natural distribution of commonsense challenges, so we only evaluate on walkthrough tuples. To
298 clarify, we also evaluate the three models on *all tuples* from **JECC** development games. The re-
299 sulted accuracies are 59.84 (BERT-Concat), 68.58 (co-match LSTM), and 68.96 (co-match BERT),
300 consistent with their ranks on **ZUCC**.

### 5.2  Human Evaluation

302 We present to the human evaluator each time a batch of tuples starting from the same observation
303 $o_t$, together with its shuffled valid actions $A_{t+1}$ and next observations $O_{t+1}$. For **JECC**, only the
304 walkthrough action $a_{t+1}$ is given. The evaluators are asked to read the start observation $o_t$ first, then
305 to align each $o \in O_{t+1}$ with an action $a \in A_{t+1}$. For each observation $o$, besides labeling the action's

Table 3: Improvement from LSTM to BERT.

| Dataset | Performance | | | $\frac{\triangle_{\text{BERT-LSTM}}}{\triangle_{\text{Human-LSTM}}}$ |
|---|---|---|---|---|
| | LSTM | BERT | Human | |
| *Multi-choice QA* | | | | |
| RACE | 50.4 | 66.5 | 94.5 | 37% |
| DREAM | 45.5 | 63.2 | 95.5 | 35% |
| *Commonsense Reasoning* | | | | |
| Abductive NLI | 50.8 | 68.6 | 91.4 | 44% |
| Cosmos QA | 44.7 | 67.6 | 94.0 | 46% |
| Our **ZUCC** | 72.3 | 72.8 | 96.4 | 2% |
| Our **JECC** | 70.0 | 74.4 | 92.0 | 20% |

alignment, the subjects are asked to answer a secondary question: whether the provided $o_t, o$ pair is sufficient for them to predict the action. If they believe there are not enough clues and their action prediction is based on a random guess, they are instructed to answer "UNK" to the second question.

We collect human predictions on 250 **ZUCC** samples and 100 **JECC** samples. The annotations are done by one of the co-authors who have experience in interactive fiction game playing (but have *not* played the development games before). The corresponding results are shown in Table 2, denoted as *Human Performance*. The human expert performs 20% higher or more compared to the machines on both datasets.

Finally, the annotators recognized 10.0% cases with insufficient clues in **ZUCC** and 17.0% in **JECC**, indicating an upper-bound of methods without access to history observations.[5]

### 5.3 Comparison to the Other Datasets

Lastly, we compare our **JECC** with the other datasets to investigate how much we can gain by merely replacing the LSTMs with pre-trained LMs like BERT for text encoding. It is to verify that the language model pre-training does not easily capture the required commonsense knowledge. When LMs contribute less, it is more likely deeper knowledge and reasoning are required so that the dataset can potentially encourage new methodology advancement. Specifically, we computed the models' relative improvement from replacing the LSTM encoders with BERT ones to measure how much knowledge BERT has captured in pre-training. Quantitatively, we calculated the ratio between the improvement from BERT encoders to the improvement of humans to LSTM models, $\triangle_{\text{BERT-LSTM}}/\triangle_{\text{Human-LSTM}}$. The ratio measures additional information (e.g., commonsense) BERT captures, compared to the full commonsense knowledge required to fill the human-machine gap.

Table 3 compares the ratios on different datasets. For a fair comparison, we use all the machine performance with co-matching style architectures. We compare to related datasets with co-matching performance available, either reported in their papers or leaderboards. These include Commonsense Reasoning datasets Abductive NLI [2] and Cosmos QA [8], and the related Multi-choice QA datasets RACE [10] and DREAM [19]. Our datasets have significantly smaller ratios, indicating that much of the required knowledge in our datasets has not been captured in BERT pre-training.

## 6 Conclusion

Interactive Fiction (IF) games encode plentiful and diverse commonsense knowledge of the physical world. In this work, we derive commonsense reasoning benchmarks **JECC** and **ZUCC** from IF games in the *Jericho Environment*. Taking the form of predicting the most likely observation when applying an action to a game state, our automatically generated benchmark covers comprehensive commonsense reasoning types such as spatial reasoning and object interaction, etc. Our experiments show that current popular neural models have limited performance compared to humans. To our best knowledge, we do not identify significant negative impacts on society resulting from this work.

---

[5]Humans can still make a correct prediction by first eliminating most irrelevant options then making a random guess.

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
