# OpenReview forum: "JECC: Commonsense Reasoning Tasks Derived fromInteractive Fictions"
_NeurIPS.cc/2021/Track/Datasets_and_Benchmarks/Round1 — Submitted to NeurIPS 2021 Datasets and Benchmarks Track (Round 1)_

### Official Review · Reviewer_U7Yy · 2021-06-30
**Common-sense Reasoning Tasks from Interactive Fictions**

**Rating:** 6
**Confidence:** 3

**Strengths:**

The main strength is that the authors are proposing a completely new approach for creating data sets for common-sense reasoning tasks. The domain of common-sense reasoning benchmarks really needs new approaches and the idea presented by the authors (to extract tasks from interactive fictions) is very interesting. The data sets created in this way do bring new quality to the problem of evaluation of common-sense reasoning.

Another strength is that strong baselines are given along with human baselines. Interesting insights are given with baselines.



**Weaknesses:**

The authors are using, at least for JECC, real walk-throughs. It is not clear whether they are really needed. Why not just use all the tuples extracted from the game? (I assume it is possible, as games are deterministic, though I'm not sure). Actually, this approach is used by the authors themselves for the ZUCC data set. The authors should show that data sets based on real walk-throughs are really superior to the ones just extracted from the games (as the latter approach is much easier).

The section on baselines is in general of good quality. I have, however, some doubts:

* more state-of-the-art Transformer models, such as RoBERTa, should be evaluated,
* it is not a weakness per se, but good results for a simple BiLSTM model (I assume it is not a pre-trained model, right?) are quite surprising and suspicious (so maybe the task is not as difficult as it seems?), i think more discussion should be devoted to this, for instance, maybe even simpler models (e.g. based on tf-idf) would obtain good results?
* it is not clear to which extent the results for BERT are affected by the fact the texts are too long, this should be assessed quantitatively (e.g. what would be the score if such cases were excluded?)

It is not clear how many annotators were employed, one or more. One is of course too few... more annotators should be used to obtain more reliable estimation of human performance.








**Additional Feedback:**

Minor remarks:
* 26: unnecessary space before the footnote mark
* 175: "walkthough"
* 127: "langauge"
* some words are lower-cased in references, e.g. "winograd", "Bert" (instead of "BERT")

**Clarity:**

The paper has some unclear moments. The discussion about reasoning beyond verification/validation (196-204) is quite unclear.

I don't think the discussion of IF games in terms of POMDPs is really needed (section 3.1). In my opinion, it could be skipped or shortened.

The paper might be sometimes hard to follow for readers without a background in reinforcement learning, for instance a short description of the Jericho environment would be helpful.

It is not clear whether human evaluation was done under the same conditions as the evaluation of neural models: in 302-303 it seems the humans saw all the possible actions, whereas 155-163 suggest that just one action $a_t^j$ is selected.

**Correctness:**

I did not find any issue here. The evaluation methods and experiment design are OK, in my opinion.

**Documentation:**

The documentation and organization of data is acceptable. Just one issue: it is not clear what is the legal status of Interactive Fiction texts used by the authors. Are they under an open license? I think this should be discussed.

**Ethics:**

I have no ethical concerns except for the issue mention in the Documentation section.

**Relation To Prior Work:**

No comments here.

**Summary And Contributions:**

The paper presents a completely new approach to creating tasks for evaluating common-sense reasoning.

The contributions are as follows:

* new common-sense reasoning data sets (ZUCC, JECC) based on interactive fictions,
* strong neural baselines for the data sets,
* human evaluation for samples of the data sets.

---

> ### Author Response · Authors · 2021-07-10
> **Look forward to your feedback!**
>
> Thanks for your detailed and constructive comments.
>
> ### Why using real walk-throughs in JECC?
> **Response**: As discussed in Line 187-188, we hope the dataset reflects a natural distribution of reasoning problems in the real world. Samples from walk-throughs correspond to the key plots in the game story thus better fit this criterion. If we use any tuples in the game, a large portion of samples will be operations on items in the inventory, which is clearly not a "natural" distribution for understanding interactive fiction.
>
> For ZUCC, sampling from only walk-throughs leads to a small dataset, so we use all tuples.
>
> ### More state-of-the-art Transformer models, such as RoBERTa, should be evaluated.
> **Response**: Thanks for the suggestion, we will add the RoBERTa results.
>
> ### It is not a weakness per se, but good results for a simple BiLSTM model (I assume it is not a pre-trained model, right?) are quite surprising and suspicious (so maybe the task is not as difficult as it seems?), i think more discussion should be devoted to this, for instance, maybe even simpler models (e.g. based on tf-idf) would obtain good results?
>
> **Response**: The LSTMs are not pre-trained. Actually, by checking the results in Table 2, the BERT models are always better than LSTM models with the same inductive biases, e.g., pair-wise inference with cross-attention (Match LSTM v.s. BERT-concat) and triple-based inference with co-matching (co-match LSTM v.s. co-match BERT). Note that BERT-siamese is a weak model that encodes the two sequences separately, so it is not an apple-to-apple comparison to the aforementioned two classes of models but just for reference. We will make it clear after Line 233.
>
> Under the same inductive biases, the improvement from LSTM to BERT is small. As discussed in Section 5.3, this confirms that solving the dataset requires additional knowledge beyond massive pre-training. This justifies the challenge of our proposed tasks.
>
> ### It is not clear to which extent the results for BERT are affected by the fact the texts are too long, this should be assessed quantitatively (e.g. what would be the score if such cases were excluded?)
> **Response**: This is a great point. The BERT-concat is affected by the long text most, while the situation for the other two BERT models is not serve. We will conduct your suggested analysis in the revised version.
>
> ### It is not clear how many annotators were employed, one or more. One is of course too few... more annotators should be used to obtain more reliable estimation of human performance.
> **Response**: Sorry for the confusion. We report the performance of a single annotator. We can add an averaged human performance by collecting data from more annotators for sure. However we are a little bit confused about whether this is necessary and we are looking for your kind advices: It seems in our field the works usually compete with top human performers only, not the average human level. And the value of the task is also estimated according to the gap between models and top human players (e.g., for the Go game machines already outperform average humans 10 years ago, but the value of building AlphaGo is to beat world champions).
>
> That says, we think our provided human performance is already good to show the value of our tasks. And unless we can find a human annotator that outperforms our co-author, or there seems not much to add by randomly collecting more human annotations. We would like to know your thoughts given our response, so we will make the revision accordingly.
>
> ### About the clarity comments
> **Response**: Thanks a lot for the suggestions. We will shorten the POMDP section and add the description of the Jericho environment.
>
> For the human study on JECC, it is under the same condition as only the walkthrough actions are asked to be labeled. For ZUCC, we did present to the annotator all the actions, or the labeling effort will be much larger. We will discuss this fact in the paper. Thanks.

---

> > ### Comment · Reviewer_U7Yy · 2021-07-14
> > **Human baseline**
> >
> > "However we are a little bit confused about whether this is necessary and we are looking for your kind advice: It seems in our field the works usually compete with top human performers only, not the average human level."
> >
> > Well, I think sometimes you compare to the average human level, sometimes to the top performer and sometimes to the committee of humans - all options are OK, I'd say. My remark was that it was simply a little bit unclear to me what you were comparing to what.

---

> > > ### Author Response · Authors · 2021-07-15
> > > **Thanks for the suggestions**
> > >
> > > Dear reviewer, thanks a lot for the suggestion. We will make this clear in the paper.
> > >
> > > Moreover, in our human study, we did first have three labelers annotating a small subset together, and find the one with top performance to annotate more data. We will add this to the paper, as well as providing the average human-level performance (on the small subset, the average performance is ~86, which clearly outperforms the neural models, too).

---

### Official Review · Reviewer_tMPj · 2021-07-03
**Predict the next observation of an interactive gameplay given the current observation and the selected action**

**Rating:** 6
**Confidence:** 3

**Strengths:**

1. Creating reasoning frameworks with natural language sentences is a relevant to evaluate models and what they can do. Although this dataset is restricted to game scripts, it’s an useful testbed for this purpose.

**Weaknesses:**

1. The authors say that in comparison to much of previous work, their dataset requires different types of reasoning to solve the task correctly, and support this claim on examples like the ones shown in Figure 1. Yet, I feel it would be good to clearly specify the *different* types of reasoning that are in the dataset and which types are required to solve each sample, maybe using human evaluation of all samples, but if not possible at least to provide some quantitative evidence. I feel that for a dataset like this it would be interesting to be able to know what ‘kinds of reasoning’ the models are not able to perform.

2. Related to the previous point, the authors discuss multi-hop reasoning and how this dataset should be useful for multi-hop systems. However, it is not clear to me how the baselines presented in S4 are multi-hop, since they seem to be roughly an LSTM+attention and BERT, and if they are not, why a multi-hop model has not been considered?

3. I’m not specially familiar with these video games, so I would appreciate it if the authors could expand on the next question. To my understanding, some of the next observations might not be driven only by commonsense or by ‘what is expected’, i.e. I would expect that the player's actions might turn out sometimes into surprise observations, traps, etc, is that correct? And if so, how would this affect the evaluation of the dataset?

4. Could the authors comment more on how this adds research value to the original Jericho environment (https://arxiv.org/pdf/1909.05398.pdf) which they consider as the starting point to create their dataset?


**Additional Feedback:**

1. Text boxes from Figure 1 are too small, IMO.
2. Line 201 mentions that ‘narratives form a whole story’ which is True, however, doesn’t this vanish with the proposed format of the dataset, since each triplet (observation, action, next_observation) seems to be handled and predicted independently from the others, is this correct?
3. In lines 169-170, 177-180 the authors discuss the problems with the drop actions and actions that have the same effects as a drop action. However, it is not clear to me how these latter actions are identified to down-sample them. Would it be possible to expand on this?
4. The Limitations  paragraph is a bit vague and little informative in its current version, so it is not really useful for the readers as it is. I suggest expanding it and discussing specific limitations.
5. In Figure 3, we see the action as a_{t+1}, but I wonder whether it should be a_{t}, as specified in other parts of the paper (e.g. line 150)?
6. In lines 278-282 the authors mention that the co-matching models perform better than the match models and say that the ‘co-matching mechanism better addressed our datasets’, but there is no explanation or hypothesis about why, would it be possible to expand on this?
7. The batch size various -> The batch size varies?
8. I had some problems understanding the evaluation of the walkthrough tuples vs all tuples. I think the source of the problem for me is that these concepts are scattered through different and non-consecutive parts of the paper (e.g. thin red lines in Figure 1, what is a walkthrough and a triplet belonging to a walkthrough, different corpora sizes for JECC and ZUCC, triplet transformation of the corpora, ...). I think it would be helpful to briefly re-collect these concepts when discussing the evaluation of walkthough vs all tuples.
9. In equation (1) (and similarly in equivalent models) H^{o,a} would correspond to the last hidden vector of the LSTM?


**Clarity:**

Overall, the paper is presented in a clear way. There are some minor aspects that can be improved such as the layout of the Figures, and a few details about the models description and the evaluation (I included my specific comments in the Additional feedback).

**Correctness:**

The methodology to create the corpus is correct, the motivation is valid and makes sense, the proposed baselines are valid and the evaluation metrics are valid as well.

**Documentation:**

The authors comment the data will be released once the paper is accepted, together with the code (MIT license). There is no mention of maintenance, but I think it should not require much.

**Ethics:**

The dataset corresponds to prompts from human interactive gameplays and thus I consider this dataset can be useful more as a testbed for models that aim to reason based on descriptions more than for developing real-world products.

I don't see major ethical concerns, but even if most of the samples are extracted from the Jericho environment, I would comment on whether users are aware that their data is being used, whether this data is anonymous, etc.


**Relation To Prior Work:**

The authors compare against different types of work, however, it is not totally clear to me in the current version why the JECC and ZUCC dataset present additional challenges to some of the previous work (for instance, the next sentence prediction datasets). Would it be possible to elaborate more on this? (see also Figure 1).

**Summary And Contributions:**

This paper presents two datasets based on human interactive gameplays from the Jericho environment and Zork Universe, where the goal is to predict the next observation based on the current observation (roughly a description of the scene the player is playing at that moment) and the action that the player took.

Among the main contributions of the paper the authors argue it is a useful dataset for developing models intended for commonsense and multi-hop reasoning.

---

> ### Author Response · Authors · 2021-07-10
> **Look forward to your feedback!**
>
> Thanks for your detailed and constructive comments. We will fix the unclear parts and typos based on your suggestion.
>
> ### I feel it would be good to clearly specify the different types of reasoning that are in the dataset and which types are required to solve each sample, maybe using human evaluation of all samples, but if not possible at least to provide some quantitative evidence. I feel that for a dataset like this it would be interesting to be able to know what ‘kinds of reasoning’ the models are not able to perform.
> **Response**: Thanks for the suggestion. There is no existing theoretical analysis or systematic ontology summarizing the commonsense knowledge types. Therefore, in the revised version, we plan to do a human study trying to summarize some knowledge types by ourselves, with examples provided.
>
> Please find our discussion regarding the multi-hop reasoning questions in the general response. Thanks.
>
>
> ### Some of the next observations might not be driven only by commonsense or by ‘what is expected’, i.e. I would expect that the player's actions might turn out sometimes into surprise observations, traps, etc, is that correct? And if so, how would this affect the evaluation of the dataset?
> **Response**: You are right that some observations are surprising ones. However, note that our task is to select the correct next observation from the distracting observations -- these surprising observations are still the more possible ones given necessary commonsense knowledge. This is reflected by our human annotation process and results -- even they are surprising observations, they are still recognized as the more likely next observations compared to the distractors.
>
> ### Could the authors comment more on how this adds research value to the original Jericho environment (https://arxiv.org/pdf/1909.05398.pdf) which they consider as the starting point to create their dataset?
> **Response**: This is a great question. As discussed in the original Jericho paper, the Jericho is proposed for several purposes: natural language understanding, commonsense understanding, and RL algorithm developments.
>
> However, as some recent papers [1] have shown, the deterministic nature of the game environment may hurt the evaluation. As when training and testing on the same game, the neural agents may just need to memorize the approximated state embeddings instead of performing real language comprehension. This makes the original metric used in Jericho, i.e., game scores, only useful for evaluating RL algorithms but less meaningful for the language comprehension purpose. Our proposed evaluation tasks add value in the NLP perspective.
>
> ### Line 201 mentions that ‘narratives form a whole story’ which is True, however, doesn’t this vanish with the proposed format of the dataset, since each triplet (observation, action, next_observation) seems to be handled and predicted independently from the others, is this correct?
> **Response**: Yes, this is our experiment setting. Currently this is ok with our tasks. As discussed in Line 314, based on human study, partial observability issues exist but do not affect human performance very much. We will make this clear in the revised version.
>
> ### In lines 169-170, 177-180 the authors discuss the problems with the drop actions and actions that have the same effects as a drop action. However, it is not clear to me how these latter actions are identified to down-sample them. Would it be possible to expand on this?
> **Response**: We have a rule-based approach because the game simulation uses specific patterns (such as *"dropped"* and *"taken"*) as consequences of these actions. We will expand on this in the updated version.
>
> ### In equation (1) (and similarly in equivalent models) H^{o,a} would correspond to the last hidden vector of the LSTM?
> **Response**: That is correct.
>
> [1] Shunyu Yao, Karthik Narasimhan, and Matthew Hausknecht. Reading and acting while blindfolded: The need for semantics in text game agents. In Proceedings of the 2021 Conference of the North American Chapter of the Association for Computational Linguistics: Human Language Technologies, pages 3097–3102, 2021.

---

### Official Review · Reviewer_ny5a · 2021-07-05

**Rating:** 5
**Confidence:** 2

**Strengths:**

The proposed problem of learning commonsense knowledge and multi-hop reasoning together from interactive fiction is interesting. The experimental results that "triple modeling" is consistently better than "textual entailment" exhibits the importance of reasoning, but it is unclear how it is related to "multi-hop".

**Weaknesses:**

The authors formulate their task as POMDP, but it is unclear how the baselines reflect the partially observable nature. It is also not clear how the “multi-hop” reasoning is modeled in these baselines. Why the comparison with human subjects is fair? Humans may not need to learn these commonsense from the IF tasks. Others see Correctness and Clarity.


Post-rebuttal

After reading the author's response, I still stand by my initial rating. The paper is vaguely written, using too many terms without careful definition. The POMDP issue is not well-justified and performance results are not clearly explained. I believe a well-done dataset paper should clearly demonstrate uniqueness, clarity, and a sufficient benchmark.

(modified: 20 Jul 2021)

**Additional Feedback:**

I think this paper can be largely improve if the authors could find an elegant way to illustrate more details in the dataset. Currently, there are too many vague high-level concepts that are ungrounded to specific examples or statistics.

**Clarity:**

The clarity of this paper can be largely improved. There are lots of vague terms that require some educated guess. For example, the authors keep mentioning "reasoning types" and "knowledge types", using these concepts to highlight the uniqueness of the proposed dataset. But it is unclear to readers what these types are. L61-65 of P2, it is written "[d]ue to the textual observation’s narrative nature, a large portion of the textual observations are not a sole statement of the action effect, but an extended narrates about what happens because of the effect", which also occurs vague to me the different of "a sole statement of the action effect" and "an extended narrates about what happens because of the effect".

In the model and the experiment sections, the authors mention they employ pretrained BERT. But it is not verbose if the pretrained model is finetuned or frozen.


**Correctness:**

The authors do not provide a justification of whether the training data is sufficient to learn the "commonsense knowledge" and the "multi-hop reasoning". I am curious why the performance on pretrained BERT does not surpass trained LSTM too much does not illustrate an insufficiency?

**Documentation:**

Details of the dataset such as how many trajectories / walkthroughs are documented. But the underlying distribution of "commonsense knowledge" is not reported.

**Ethics:**

No ethical issues found.

**Relation To Prior Work:**

I am not quite familiar with this literature. As mentioned in weakness, the evaluation of relation to prior work is substantially hindered by the vagueness in some key concepts.

**Summary And Contributions:**

This paper proposes two Interactive fiction tasks in which the learning of commonsense knowledge and multi-hop reasoning are important. The authors claim that in these tasks a model would need to learn "various types" of commonsense in the form of world model learning. They propose several baselines based on BiLSTM and BERT, and show the gap between human performance.

---

> ### Author Response · Authors · 2021-07-10
> **Look forward to your feedback!**
>
> Thanks again for your constructive comments. While the comments are inspiring and we learned a lot from them, we feel it is unfair to count the raised correctness issue as weaknesses. To justify this, please allow us to start with the response to the "correctness" comment and clarify our motivation.
>
> ### About Correctness: The authors do not provide a justification of whether the training data is sufficient to learn the "commonsense knowledge" and the "multi-hop reasoning". I am curious why the performance on pretrained BERT does not surpass trained LSTM too much does not illustrate an insufficiency?
>
> **Response**: We understand the reviewer's opinion. However, this is usually not considered as a weakness as a commonsense dataset paper. Most of the previous datasets in Section 2 are not aiming at proposing a training dataset that contains sufficient required knowledge. Instead, they aim at providing a benchmark for future work to equip the models with additional commonsense knowledge.
>
> In short, most of the commonsense reasoning datasets hope to provide a dataset, the training data of which is **insufficient** to capture the knowledge, but the task can be potentially resolved with knowledge acquired from other resources. This is the same case for our work.
>
> However, although previous datasets successfully posed new challenges, many studies found that they can be easily "conquered" with pretrained models like BERT. But despite the success of BERT on these datasets and the fact that BERT captures certain commonsense knowledge as demonstrated by a series of probing and prompting approaches, there is still a research question: whether BERT is already sufficient for daily-used commonsense.
>
> Many people guessed the answer is "No". And **this is one important motivation of our work**: It is a well-accepted fact that understanding the IF games has considerable coverage of commonsense knowledge (e.g., the discussion on Page 2 of [1], also see examples in the appendix of [2]). Then based on our constructed tasks, if BERT fails to improve much, that reflects humans' commonsense is not simply captured by massive pre-training (similarly, BERT will fail to improve a lot if more reasoning skills are required). And this is exactly our observation: since BERT does not have a clear advantage in covering necessary commonsense in our tasks, its improvement over LSTM models is limited.
>
> ### The authors formulate their task as POMDP, but it is unclear how the baselines reflect the partially observable nature.
> **Response**: Our baselines do not address the partial observability nature. As discussed in Line 314, based on human study, partial observability issues exist but do not affect human performance very much. We will make this clear in the revised version.
>
> ### Why the comparison with human subjects is fair? Humans may not need to learn these commonsense from the IF tasks.
> **Response**: Please refer to our response to the correctness issue. The comparison is fair in the following perspective: considering the motivations of the commonsense dataset construction we discussed, we are evaluating the commonsense knowledge captured by BERT and humans on the IF tasks. The commonsense should not be solely learned from our tasks. Instead, our training data is used to help the BERT models understand the task and learn to use the commonsense captured in the pretrained models.
>
> ### The authors keep mentioning "reasoning types" and "knowledge types", using these concepts to highlight the uniqueness of the proposed dataset. But it is unclear to readers what these types are.
> **Response**: There is no existing theoretical analysis or systematic ontology summarizing the commonsense knowledge types. Thus we believe there is an advantage of our work -- starting with IF games that are believed to require commonsense, we can ensure the coverage of certain commonsense types even without a theoretical categorization. This is an alternative approach comparing to the related work, which first decides a single type of knowledge then creates a dataset from scratch accordingly.
>
> In the revised version we plan to summarize our own knowledge type categorization with examples provided.
>
> ### Difference between "a sole statement of the action effect" and "an extended narrates about what happens because of the effect".
> **Response**: Please refer to the first part of our general response.
>
> ### In the model and the experiment sections, the authors mention they employ pretrained BERT. But it is not verbose if the pretrained model is finetuned or frozen.
> **Response**: the BERT models are finetuned on our training data.
>
> [1] Matthew Hausknecht, Prithviraj Ammanabrolu, Marc-Alexandre Côté, and Xingdi Yuan. Interactive fiction games: A colossal adventure. arXiv preprint arXiv:1909.05398, 2019.
>
> [2] Shunyu Yao, Rohan Rao, Matthew Hausknecht, and Karthik Narasimhan. Keep calm and explore: Language models for action generation in text-based games. EMNLP 2020.

---

### Author Response · Authors · 2021-07-10
**General response regarding the multi-hop reasoning and corresponding baselines**

We thank all reviewers for their constructive comments. We try to answer some common questions regarding multi-hop here.

### Examples of multi-hop reasoning in our tasks

To explain the question about multi-hop baselines, we hope to first clarify the multi-hop challenges existing in our dataset. (We hope this clarification can also help address the issue asked by R1 about "an extended narrates about what happens because of the effect".)

Multi-hop reasoning is a special case of deduction. Taking two-hop as an example, formally, with a knowledge resource $D$, a fact $fact_c$ can be inferred from an input fact fact_a, if we have $f(fact_a, x_1 \in D) => fact_b$ and $f(fact_b, x_2 \in D) =>fact_c$. Here $f(\cdot,\cdot)$ indicates a (soft) rule over two facts.

In our dataset, a corresponding 2-hop inference example from $fact_a$ to $fact_c$ is:

$fact_a$ $--$ ($o_t$="Land of Shadow: You are in a land of dark shadows and shallow hills, which stretch out in all directions. To the west, the land dips sharply.", $a_t$="north")

$fact_c$ $--$ $o_{t+1}$="Land of Shadow: You are standing atop a steep cliff, looking west over a vast ocean. Far below, the surf pounds at a sandy beach. To the south and east are rolling hills filled with eerie shadows. A path cut into the face of the cliff descends toward the beach. To the north is a tall stone wall, which ends at the cliff edge."

Based on commonsense knowledge about spatial relations, it can immediately infer a consequence $fact_b$="north of the land of shadow". That is, the effect of taking action $a_t$ at $o_t$. While $fact_c$ can be inferred from $fact_b$ since "to the north is a tall stone wall, which ends at the cliff edge" indicates this is the "north end of the 'land of shadow'". Therefore, for this example, inferring $fact_c$ from $fact_a$ is a two-step process hopping at $fact_b$, with each step requiring different types of knowledge.

Actually the aforementioned two-step process happens to most of the tuples, but the second step of reasoning can usually be easy to resolve for humans.

We will make this clear and provide examples in the revised version.

### About the multi-hop baselines

As shown in the previous example, our multi-hop cases pose some challenges since the facts are not explicitly provided but are described with texts. For such task of multi-hop reasoning over multi-domain texts, there is not a well-accepted explicit multi-hop reasoning model, to the best of our knowledge. Existing state-of-the-arts usually rely on attention models to implicitly conduct such reasoning. The co-matching style model has been shown to have a good multi-hop modeling capability, such as taking orders of facts into consideration to improve performance, as studied in the reference [21]. Therefore we take the co-matching models as implicit multi-hop baselines. We will make this clear in the revised version.

---

### Decision · Program_Chairs · 2021-07-27

**Decision:**

Reject

**Comment:**

The proposed dataset interesting and unique. However, there are also still significant limitations rased by the reviewers that should be addressed before accepting the paper. Also, the presentation of the paper should be substantially improved, and the performance results should be better explained. After discussion with multiple area chairs, we feel that the paper is not yet ready for publication. Perhaps the remaining issues could be addressed before the second round of this track.